# *idh-1* neomorphic mutation confers sensitivity to vitamin B12 in *Caenorhabditis elegans*

Olga Ponomarova[1,2], Alyxandra N Starbard[1], Alexandra Belfi[3], Amanda V Anderson[2], Meera V Sundaram[3], Albertha JM Walhout[1]

In humans, a neomorphic isocitrate dehydrogenase mutation (*idh-1neo*) causes increased levels of cellular D-2-hydroxyglutarate (D-2HG), a proposed oncometabolite. However, the physiological effects of increased D-2HG and whether additional metabolic changes occur in the presence of an *idh-1neo* mutation are not well understood. We created a *Caenorhabditis elegans* model to study the effects of the *idh-1neo* mutation in a whole animal. Comparing the phenotypes exhibited by the *idh-1neo* to Δ*dhgd-1* (D-2HG dehydrogenase) mutant animals, which also accumulate D-2HG, we identified a specific vitamin B12 diet-dependent vulnerability in *idh-1neo* mutant animals that leads to increased embryonic lethality. Through a genetic screen, we found that impairment of the glycine cleavage system, which generates one-carbon donor units, exacerbates this phenotype. In addition, supplementation with alternate sources of one-carbon donors suppresses the lethal phenotype. Our results indicate that the *idh-1neo* mutation imposes a heightened dependency on the one-carbon pool and provides a further understanding of how this oncogenic mutation rewires cellular metabolism.

## Introduction

Increased levels of D-2-hydroxyglutarate (D-2HG), a metabolite derived from the structurally similar hub metabolite alpha-ketoglutarate (αKG), are associated with multiple disorders, indicating that tight regulation of D-2HG is important (1, 2). For instance, D-2-hydroxyglutaric aciduria, a rare inborn error of metabolism, is associated with elevated D-2HG levels due to loss-of-function mutations in the D-2HG dehydrogenase enzyme (3). This inborn error of metabolism often results in neurological dysfunctions and delayed development. Previously, we found that loss of the *Caenorhabditis elegans* D-2HG dehydrogenase (*dhgd-1*) causes a high rate of embryonic lethality due to reduced ketone body production

(4). Additionally, we found that *dhgd-1* activity is necessary for the regulation of the propionate shunt, an alternate vitamin B12-independent breakdown pathway for this short chain fatty acid (5). In this shunt, the enzymes DHGD-1 and HPHD-1 are coupled via D-2HG metabolism: HPHD-1 transfers a hydride from 3HP to α-ketoglutarate (αKG), producing D-2HG, whereas DHGD-1 oxidizes D-2HG back to αKG (4, 5) (Fig 1A). The propionate shunt is transcriptionally repressed in the presence of vitamin B12 (6). Vitamin B12 rescues the embryonic lethality of Δ*dhgd-1* mutants by generating energy via the canonical propionate degradation pathway, alleviating the need for ketone bodies to distribute an energy source across tissues (4) (Fig 1A).

D-2HG is also known as an oncometabolite and is linked to various cancers. D-2HG accumulates due to neomorphic mutations in either one of two isocitrate dehydrogenase (IDH) enzymes (IDH1 and IDH2). These mutations primarily affect catalytic arginine residues (7, 8, 9) and are associated with the development of cancers such as glioma, cholangiocarcinoma, and AML (10, 11, 12).

Neomorphic mutations in IDH1 and IDH2 enzymes lead to abnormal D-2HG production from αKG (13), thereby disrupting cell function. Effects of D-2HG are multifaceted and can drive cancer development by several different mechanisms (1, 14). D-2HG acts as a potent competitive inhibitor of αKG-dependent enzymes, including histone demethylases and hypoxia-inducible factor prolyl hydroxylase, often leading to dysregulated oncogene expression (15). Abnormal D-2HG production also disturbs the balance between NADPH and NADP+, crucial for cellular redox equilibrium (16). This disruption can cause oxidative stress, leading to DNA damage. High levels of D-2HG have also been shown to inhibit succinate dehydrogenase (17) and αKG-dependent transaminases (16), disrupt chromosomal topology (18), and activate the mTOR pathway (19). D-2HG also affects the immune system, particularly T cells, potentially creating a tumor-friendly environment by suppressing the immune response. Malignant cells with IDH mutations release D-2HG, which can suppress T-cell function by inhibiting lactate dehydrogenase and disrupt other metabolic pathways (20, 21, 22). Increased levels of D-2HG caused by the inhibition of D-2-hydroxyglutarate dehydrogenase activity have also been associated

[1]Department of Systems Biology, University of Massachusetts Chan Medical School, Worcester, MA, USA   [2]Department of Biochemistry and Molecular Biology, University of New Mexico School of Medicine, Albuquerque, NM, USA   [3]Department of Genetics, University of Pennsylvania Perelman School of Medicine, Philadelphia, PA, USA

Correspondence: oponomarova@salud.unm.edu; marian.walhout@umassmed.edu

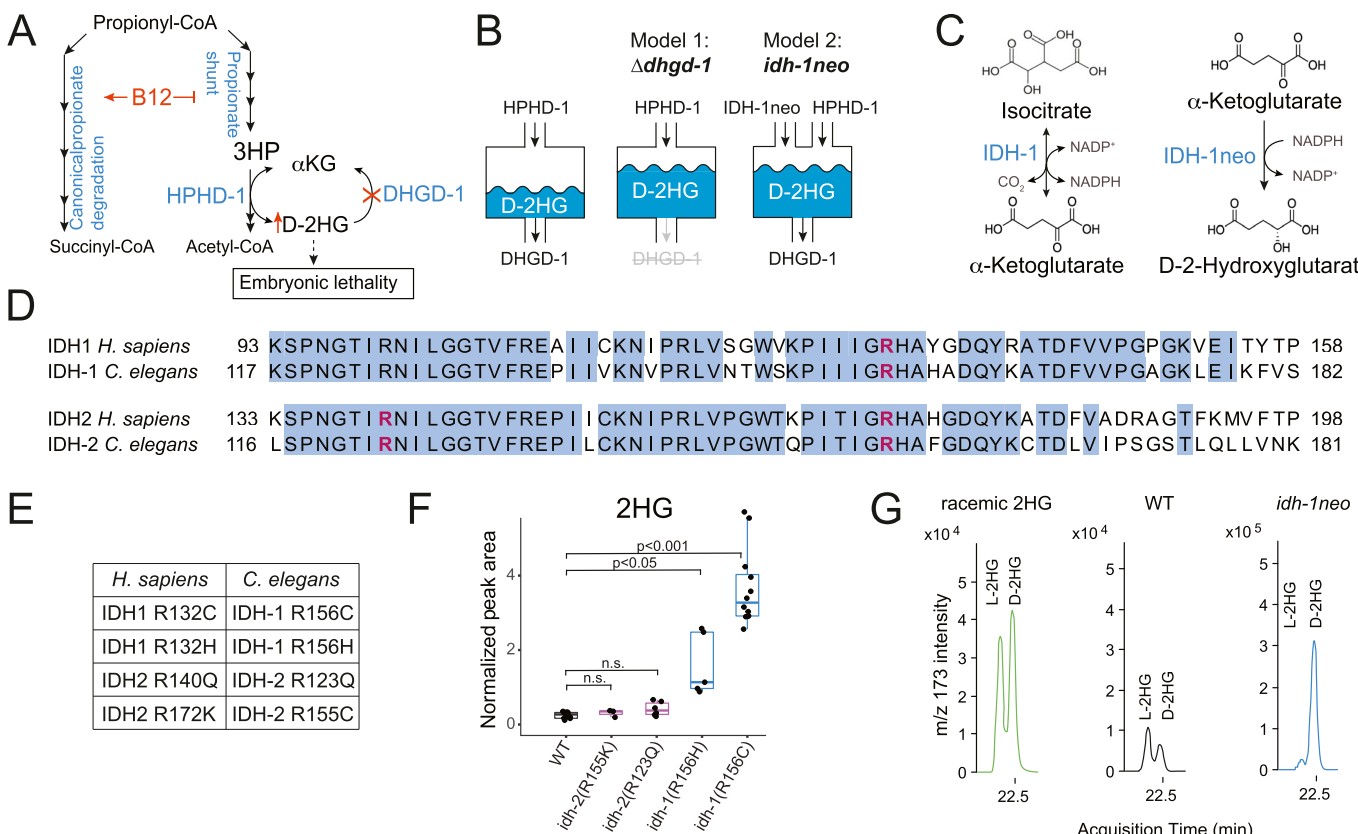

**Figure 1. Introduction of exogenous *idh-1* with neomorphic mutation leads to D-2HG accumulation in *C. elegans*.**
**(A)** Interplay between D-2-hydroxyglutarate dehydrogenase *dhgd-1*, vitamin B12, and the propionate shunt pathway (4). **(B)** Models of D-2HG accumulation. Mutation in D-2-hydroxyglutarate dehydrogenase *dhgd-1* prevents D-2HG recycling (Model 1). Neomorphic mutation in isocitrate dehydrogenase 1 (*idh-1neo*) creates a new source of D-2HG (Model 2). **(C)** Reactions catalyzed by WT and neomorphic IDH. **(D)** Protein sequences of IDH1 and IDH2 from human and *C. elegans*. Alignment was performed using Clustal Omega software. Blue color highlights conserved residues. Arginine residues homologous to those that typically mutate in human cancers are shown in pink. **(E)** Amino acid substitutions in *C. elegans* IDH-1 and IDH-2 that correspond to common human cancer-associated mutations. **(F)** GC-MS quantification of 2HG (D- and L-2HG) in *C. elegans* expressing neomorphic *idh-1* and *idh-2*. Boxplot midline represents median of independent biological replicates (dots). **(G)** *idh-1neo* animals predominantly accumulate the D isoform of 2HG. D- and L-2HG enantiomers were measured by GC-MS after chiral derivatization.

with different cancers (23, 24, 25). Whereas many effects of D-2HG are well-documented, the complete implications of dysregulated D-2HG metabolism remain unclear. Its versatile effects range from supporting oncometabolism to causing developmental and psychomotor defects in D-2-hydroxyglutaric aciduria patients. Understanding the diverse toxic effects of D-2HG is crucial for unraveling disease progression mechanisms and developing new treatments.

To gain a better understanding of how D-2HG impacts cellular metabolic function, we generated *C. elegans idh-1neo* mutant animals to use as a comparative model for studying the effects of increased D-2HG levels. We find that whereas some shifts in metabolism are shared with what we found previously in our studies of Δ*dhgd-1* mutant animals (4), differences exist. These differences led us to uncover a unique diet-dependent, vitamin B12-induced vulnerability in *idh-1neo* mutant animals. Whereas vitamin B12 rescues embryonic lethality in Δ*dhgd-1* mutant animals, it exacerbates lethality of *idh-1neo* mutant animals. We find that this difference is due to decreased one carbon metabolism in *idh-1neo* mutant animals. Overall, our results provide a further

understanding of how the *idh-1neo* oncogenic mutation may rewire cellular metabolism.

# Results

## *C. elegans* with neomorphic *idh-1* mutation accumulate D-2HG

We previously found that when the function of the *C. elegans* D-2HG dehydrogenase, *dhgd-1*, is disrupted, there is a marked increase in D-2HG levels in the animals (Fig 1B, Model 1) (4). Seeking to further understand the metabolic implications of D-2HG accumulation, we aimed to increase D-2HG levels through a distinct mechanism; by introducing an exogenous D-2HG-producing enzyme (Fig 1B, Model 2). Neomorphic mutations in IDH, whether cytosolic (IDH1) or mitochondrial (IDH2), alter enzyme function; rather than converting isocitrate to αKG, these mutant enzymes convert αKG to D-2HG (Fig 1C). We separately introduced four *idh* alleles with missense mutations that mirror human neomorphic variants into *C. elegans*

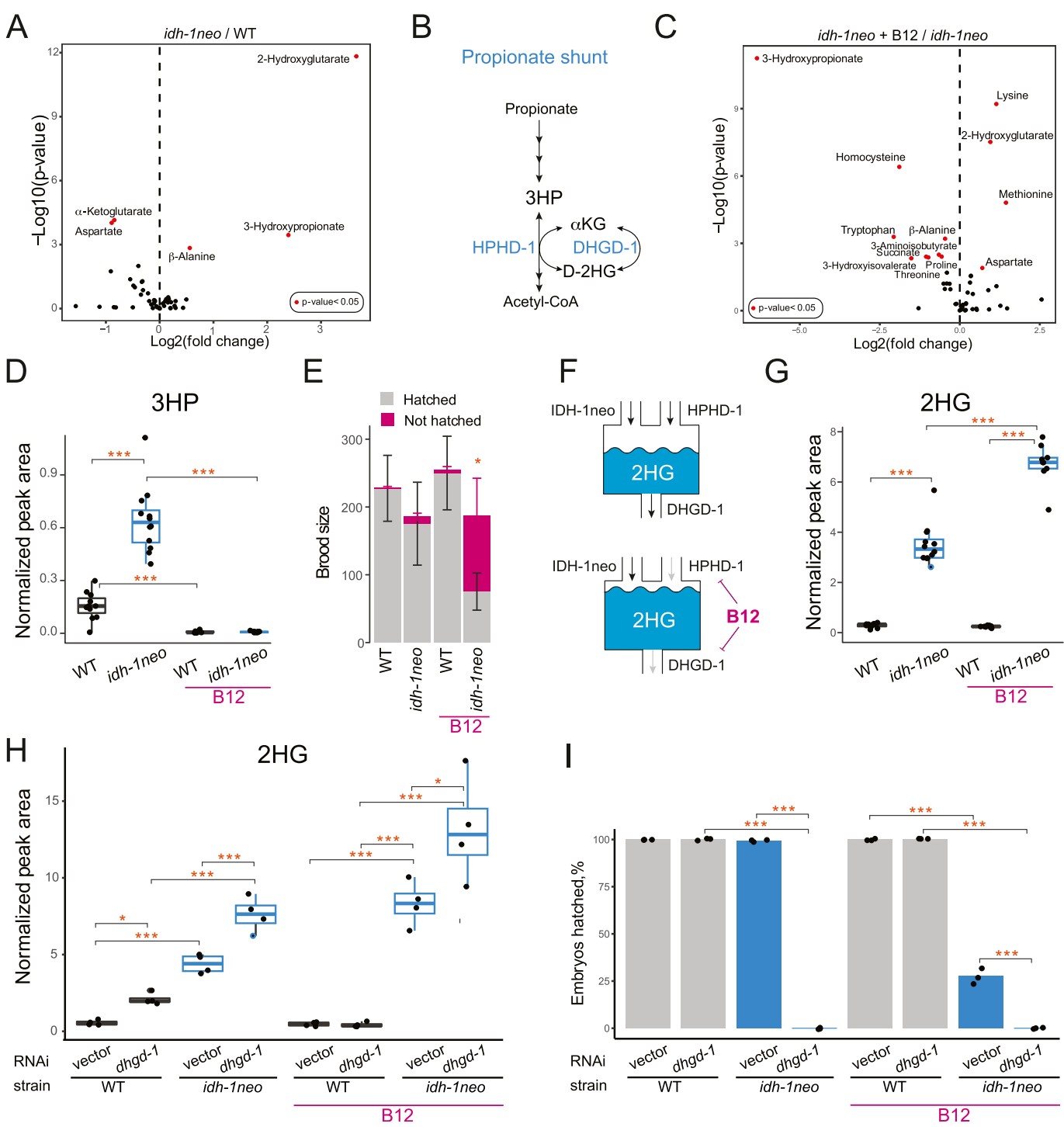

**Figure 2. Vitamin B12 increases D-2HG levels and induces embryonic lethality in *idh-1neo C. elegans*.**
**(A)** GC-MS profiling of metabolic changes in *idh-1neo C. elegans* compared with WT animals. *P*-values are Benjamini–Hochberg adjusted. Alanine is a proteinogenic amino acid whereas *β*-alanine is a non-proteinogenic amino acid intermediate in pyrimidine, aspartate, and propionate metabolism. **(B)** Schematic of the propionate degradation shunt pathway. **(C)** GC-MS profiling of metabolic changes in *idh-1neo C. elegans* supplemented with vitamin B12 compared with non-supplemented *idh-1neo* animals. *P*-values are Benjamini–Hochberg adjusted. **(D)** 3HP abundance in *idh-1neo* mutants with and without supplemented vitamin B12. **(E)** Brood size and hatching rate of *idh-1neo* mutants. Bars represent mean and SD of three biological replicates. **(F)** Schematic of predicted IDH-1neo, HPHD-1, and DHGD-1 contributions to D-2HG accumulation. IDH-1neo is an introduced and HPHD-1 is an endogenous source of D-2HG. DHGD-1 recycles D-2HG by converting it to *α*KG. In *idh-1neo* animals, vitamin B12 causes an increase in D-2HG accumulation by repressing the expression of *dhgd-1*, which encodes the enzyme that converts D-2HG into *α*KG in the propionate shunt. Vitamin B12 also represses the expression of *hphd-1*, which is the enzyme that generates D-2HG in the propionate shunt. **(G)** 2HG abundance in *idh-1neo* mutants with and without supplemented vitamin B12. **(H)** 2HG abundance in *idh-1neo* mutants upon RNAi of *dhgd-1* with and without supplemented vitamin B12. **(I)** Embryonic lethality of

(Fig 1D and E), whereas keeping the endogenous WT *idh* genes intact. Using these mutant strains, we assessed 2HG (D- and L-2HG) accumulation in the animals by gas chromatography—mass spectrometry (GC-MS). Whereas neither *idh-2* allele resulted in 2HG accumulation, animals harboring an *idh-1* alleles did show an increased accumulation of 2HG, with the highest levels found in animals expressing the *idh-1*(R156C) allele (Fig 1F). Approximately 50% of *idh-1*(R156C) animals also showed developmental abnormalities, including dilations in the excretory system, larval lethality, and a smaller proportion of embryonic lethality, suggesting systemic disruptions to animal physiology (Fig S1A and B and Table S1). Therefore, we chose this strain for detailed exploration, and hereafter refer to it as *idh-1neo*.

2HG exists as either the D-2HG or L-2HG enantiomer, and neomorphic IDH mutations specifically cause production of D-2HG (27). Using a specific derivatization technique to distinguish 2HG enantiomers (28), we confirmed that *idh-1neo* animals accumulate D-2HG (Fig 1G). These combined results show that we have generated a new model for D-2HG accumulation in *C. elegans*, distinct from that in Δ*dhgd-1* animals (4). Using these two models, we went on to further understand the metabolic implications of D-2HG accumulation.

### Vitamin B12 supplementation increases D-2HG levels and causes embryonic lethality in *idh-1neo* mutants

We next investigated whether metabolites other than D-2HG change in abundance in *idh-1neo* animals. GC-MS metabolomics revealed that, much like Δ*dhgd-1* animals, *idh-1neo* animals exhibited elevated levels of 3-hydroxypropionate (3HP) and β-alanine, along with reduced levels of αKG and aspartate (Fig 2A) (4).

In Δ*dhgd-1* mutants, 3HP accumulates because D-2HG inhibits HPHD-1, an enzyme that produces D-2HG whereas oxidizing 3HP in the propionate shunt pathway (4, 29) (Fig 2B). To determine if 3HP accumulation in *idh-1neo* also originates from this shunt pathway, we supplemented the animals with vitamin B12. Vitamin B12 transcriptionally inhibits the propionate shunt pathway whereas promoting the activity of the canonical, B12-dependent propionate degradation pathway (5, 6) (Fig 1A). We reasoned that if suppressed HPHD-1 activity is the cause of 3HP accumulation, then inhibiting the entire shunt pathway should prevent it. Indeed, vitamin B12 supplementation led to reduced 3HP levels in *idh-1neo* animals, which also occurred in Δ*dhgd-1* mutant animals (Fig 2C and D). Interestingly, in contrast to Δ*dhgd-1* mutant animals, vitamin B12 supplementation significantly increased the rate of embryonic but not larval lethality in the F1 generation of *idh-1neo* animals (Fig 2E and Table S1) (4). We hypothesized that because of vitamin B12 suppresses the expression of shunt pathway genes, including *dhgd-1*, its supplementation may hinder DHGD-1 dependent D-2HG recycling, thereby further elevating D-2HG levels in *idh-1neo* animals (Fig 2F). Indeed, adding vitamin B12 to the diet of the *idh-1neo* significantly increased their D-2HG levels (Figs 2G and S2). To test this hypothesis further, we asked if suppressing *dhgd-1* expression

would elevate D-2HG in *idh-1neo* animals. As predicted, *dhgd-1* RNAi was sufficient to drive further increase in D-2HG levels in *idh-1neo* animals (Fig 2H). Importantly, *dhgd-1* RNAi also led to 100% penetrant embryonic lethality among the F1 generation of *idh-1neo* animals (Fig 2I). In contrast, *hphd-1* RNAi did not cause embryonic lethality, further demonstrating that lack of 3HP degradation is not linked to this phenotype (Fig S3) (4).

The opposite response to vitamin B12 supplementation highlighted key differences between the two models of D-2HG accumulation. The embryonic lethality observed in Δ*dhgd-1* animals arises from a lack of energy source (ketone bodies) and can be rescued by vitamin B12, which activates an alternative energy production pathway (4). In contrast, embryonic lethality of *idh-1neo* animals is induced by vitamin B12 and cannot be mitigated by ketone body supplementation (Fig S4). We therefore conclude that *idh-1neo* mutation causes embryonic lethality through a different molecular mechanism.

### Knockdown of the glycine cleavage system (GCS) exacerbates lethality of *idh-1neo* animals supplemented with vitamin B12

To identify the molecular mechanism underlying the lethality of *idh-1neo* animals in the presence of vitamin B12, we conducted a reverse genetic screen. We used an RNAi library targeting 2,104 predicted metabolic genes (30) to identify those that are essential for *idh-1neo* animals but not required for WT *C. elegans* survival in the presence of vitamin B12 (Fig 3A). The screen identified five metabolic genes whose depletion is specifically lethal to *idh-1neo* animals (Figs 3B and S5). Among these, two genes, *T04A8.7* and *W07E11.1*, encode a glycogen branching enzyme and glutamate synthase, respectively. The other three identified genes—*gldc-1*, *gcst-1*, and *gcsh-1*—all belonging to the GCS (31). Two other GCS genes, *gcsh-2* and *dld-1* were not identified as "hits." *gcsh-2* is associated with the same reaction as *gcsh-1*, indicating that the latter encodes an active enzyme (31). *dld-1* functions in other metabolic processes, particularly in lactate/pyruvate metabolism, and confers embryonic lethality when knocked down in WT animals (32). Given the strong enrichment for GCS in our screen results, we next considered possible connections between GCS, vitamin B12, and *idh-1neo*.

### *idh-1neo* mutation confers sensitivity to perturbations of one-carbon metabolism

The GCS breaks down glycine, thereby generating ammonia, carbon dioxide, and reducing NAD+ to NADH, whereas also methylating tetrahydrofolate, a one-carbon (1C) unit donor used for different biosynthetic reactions (Fig 4A). 1C metabolism, similar to the canonical propionate breakdown pathway, also depends on vitamin B12: in the methionine/S-adenosylmethionine (Met/SAM) cycle, METR-1 (methionine synthase) methylates homocysteine to regenerate methionine using vitamin B12 as a cofactor. The Met/SAM cycle utilises 1C units provided by the enzyme methylenetetrahydrofolate

---

*idh-1neo* mutants upon RNAi of *dhgd-1* with and without supplemented vitamin B12. ***$P < 0.001$. All panels show data for *idh-1neo* animals on a diet of *E. coli* OP50 or RNAi competent *E. coli* OP50 (xu363). **(D, G, H)** Boxplot midline in panels (D, G, H) represents median of independent biological replicates (dots).

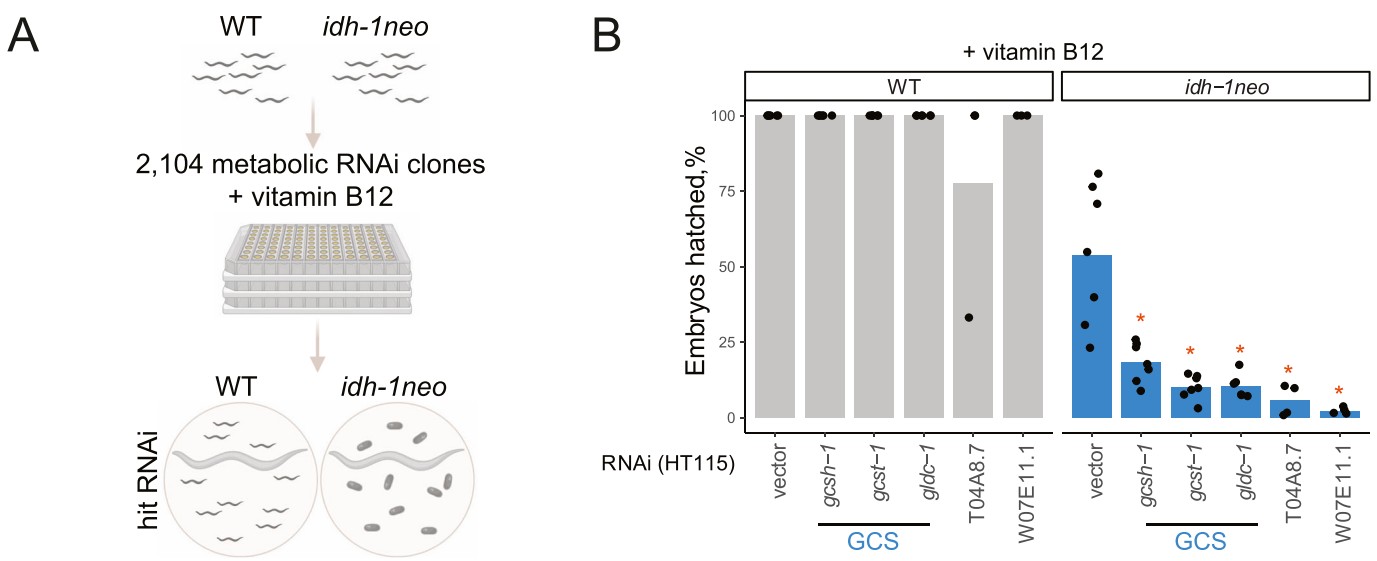

**Figure 3. Metabolic RNAi screen for synthetic lethality with *idh-1neo*.**
**(A)** Experimental design to screen metabolic RNAi library for synthetic (embryonic) lethal interactions with *idh-1neo*. Created with BioRender.com. **(B)** "Hits" identified in metabolic RNAi library screen. *$P$-value < 0.05, compared with vector control.

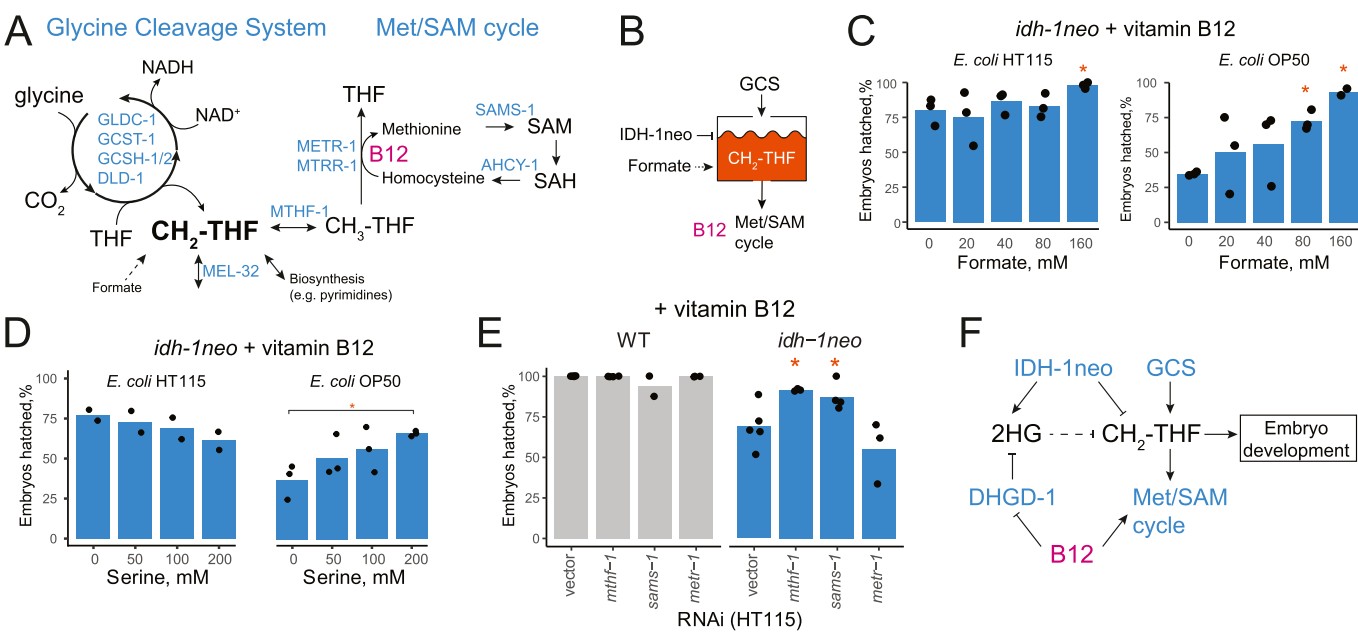

**Figure 4. *idh-1neo* animals are sensitive to perturbing one-carbon metabolism.**
**(A)** Glycine cleavage system contributes to a pool of one carbon units, and methionine/S-adenosylmethionine (SAM) cycle draws from this pool. **(B)** Hypothesized interaction of *idh-1neo* with vitamin B12 and glycine cleavage system via 1C pool. Glycine cleavage system or supplemented formate replenish 1C pool. Vitamin B12 depletes it by promoting Met/SAM cycle activity. **(C)** Embryonic lethality of *idh-1neo* on vitamin B12 is rescued by supplementing formate on diets of *E. coli* OP50 and HT115. *$P$-value < 0.05, compared with no supplement condition. **(D)** Embryonic lethality of *idh-1neo* on vitamin B12 is rescued by supplementing serine on a diet of *E. coli* OP50. *$P$-value < 0.05, compared with no supplement condition. **(E)** Suppressing Met/SAM cycle activity rescues embryonic lethality of *idh-1neo* supplemented with vitamin B12. *$P$-value < 0.05, compared with vector control. **(F)** Model for the *idh-1neo* interaction with 1C metabolism.

reductase MTHF-1 (33) (Fig 4A). Both the GCS and the Met/SAM cycle influence the 1C pool of methylene tetrahydrofolate: GCS contributes to its synthesis, whereas the Met/SAM cycle used it. Therefore, we hypothesized that *idh-1neo* animals are sensitive to depletion of the 1C pool (Fig 4B). To test this hypothesis, we supplemented B12-treated *idh-1neo* animals with formate, an alternative 1C donor

(34). The highest doses of supplemented formate somewhat slowed the development of P0 animals but restored the survival of *idh-1neo* embryos to WT levels on a regular diet of *E. coli* OP50 as well as the diet of RNAi-competent *E. coli* HT115 (Figs 4C and S6A and B). The alternative 1C donor, serine, also rescued embryonic lethality of *idh-1neo* animals, but only when fed an *E. coli* OP50 diet (Figs 4D and S7A

and B). Furthermore, we posited that if vitamin B12 induces lethality in *idh-1neo* animals by depleting the 1C pool via its utilization in the Met/SAM cycle, then suppressing Met/SAM cycle genes in *idh-1neo* should prevent this depletion and restore availability of 1C units for other reactions (Fig 4A). Indeed, RNAi depletion of *mthf-1* and *sams-1* (SAM synthetase) rescued the embryonic lethality of *idh-1neo* animals supplemented with vitamin B12 (Figs 4E and S8). These findings demonstrate that lack of 1C units contributes to the embryonic lethality observed in *idh-1neo* animals.

# Discussion

By comparing two models of D-2HG accumulation in *C. elegans*, we have gained deeper insight into the metabolic perturbations caused by D-2HG in a whole animal. Similarities between Δ*dhgd-1* and *idh-1neo* include the perturbed function of the propionate shunt enzyme HPHD-1, evident from an increase in levels of its substrate 3HP. Other similarities include elevated *β*-alanine and reduced *α*KG. The differences in the metabolic phenotypes of the two models include changes in lysine, 2-aminoadipate, and glutarate levels, and can be linked to the compartmentalization of D-2HG production and the different subcellular origins of D-2HG: DHGD-1 recycles D-2HG produced by HPHD-1 in mitochondria, whereas *IDH-1neo* generates D-2HG in the cytosol. DHGD-1 dysfunction is thus more likely to affect mitochondrial enzymes whereas IDH-1neo may have a stronger impact on cytosolic metabolism. Consistent with this theory, 3HP levels, indicative of HPHD-1 inhibition, are several-fold higher in Δ*dhgd-1* mutants than in *idh-1neo* animals. In further support of subcellular stratification, mitochondrial lysine degradation pathway intermediates (lysine and 2-aminoadipate) change levels in Δ*dhgd-1* mutants, but not in *idh-1neo* animals (Fig 2A). These lysine levels, however, become perturbed in *idh-1neo* when vitamin B12, a transcriptional suppressor of *dhgd-1*, is supplemented (Fig 2C).

1C units in the form of methylated tetrahydrofolate are essential metabolic intermediates used for nucleotide biosynthesis and various methylation reactions (35). A lack of these building blocks results in embryonic lethality (34). Formate, a one-carbon donor exchanged between mitochondria and cytosol, has been demonstrated to rescue these detrimental effects (36). Our results show that *idh-1neo C. elegans* rely on GCS to supply one-carbon units. We propose that the metabolic rewiring caused by the *idh-1neo* mutation reduces the availability of methylated tetrahydrofolate. This limitation, in turn, causes sensitivity of *idh-1neo* to vitamin B12 and GCS knockdown, both of which can drain the 1C pool (Fig 4F). We propose that a lack of 1C units in *idh-1neo* can impede pyrimidine biosynthesis via thymidylate synthase *tyms-1*, which uses 1C units to generate dTMP. Supporting this hypothesis, RNAi of *tyms-1* causes embryonic lethality (37, 38, 39). WT *C. elegans* can generate 1C via cytosolic serine hydroxymethyltransferase MEL-32, whose loss causes embryonic lethality (31, 33, 40). Why would the MEL-32 route for 1C unit generation not be available in *idh-1neo* animals? One possibility is inhibition of this pathway through accumulated D-2HG. The phosphoglycerate dehydrogenase C31C9.2 functions upstream of MEL-32 (31), and its human ortholog PHGDH was found

to produce D-2HG (41). A recent study demonstrated that D-2HG accumulation in Δ*dhgd-1* animals suppresses the activity of the D-2HG-producing enzyme HPHD-1 (29). A similar mechanism of end-product inhibition could cause the excess D-2HG produced by *idh-1neo* to suppress C31C9.2 activity, limiting the downstream generation of 1C by MEL-32.

Overall, our results uncover metabolic perturbations induced by the *idh-1neo* mutation and highlight the differences in the pathogenicity mechanism of *idh-1neo* and Δ*dhgd-1* models. Whereas both mutants accumulate D-2HG and incur embryonic lethality, the Δ*dhgd-1* phenotype is caused by a lack of ketone bodies, while *idh-1neo* suffers from a 1C deficiency. Comparing the two models offers a unique tool for mechanistic insight. These findings may help navigate metabolic reprogramming that occurs in IDH-driven oncogenic transformations. Whereas our results have focused on how the neomorphic *idh-1* mutation affects the developing embryo, proliferating cancer cells also have been shown to have increased demand for 1C units, for instance, to synthesize nucleosides (34, 35). Thus, we can speculate that cancers with mutated IDH1 may be increasingly sensitive to depletion of the 1C pool, also. Future studies may explore 1C metabolism as a potential target in the therapy of cancers with the IDH1mutation.

# Materials and Methods

### Bacterial strains

*E. coli* HT115, *E. coli* OP50 (xu363) (42), and *E. coli* OP50 from Caenorhabditis Genetics Center (CGC) were cultured overnight in Luria-Bertani Broth (Miller) at 37°C, plated, and incubated overnight on assay plates before adding *C. elegans* larvae. For RNAi experiments, *E. coli* HT115 was used, and assay plates were supplemented with *μ*g/ml 50 ampicillin and 1 mM isopropyl *β*-d-1-thiogalactopyranoside (IPTG).

### *C. elegans* cultures

*C. elegans* strains (Table 1) were maintained at 20°C on nematode growth medium (NGM) seeded with *E. coli* HT115 or OP50. All experiments were performed using an *E. coli* OP50 diet, unless specified otherwise. Supplements were added to NGM agar as specified. Vitamin B12 (adenosylcobalamin) was used at 64 nM throughout. N2 strain was obtained from CGC and mutant strains were constructed as described below.

### Constructing *C. elegans* strains

Transgenic *C. elegans* strains with neomorphic mutations in *idh-1* and *idh-2* were created by inserting a mutated gene in an intergenic region on chromosome II at position 8420158..8420158 using Mos1-mediated single copy insertion (MosSCI) technique (44). We used expression of an added allele to ensure that endogenous *idh-1* remains functional because WT IDH1 activity was demonstrated to be necessary for efficient D-2HG production in cells with monoallelic neomorph mutations of IDH1 (45). WT *idh* genes, together

**Table 1.** *C. elegans* strains used in this study.

| Designation | Genotype | Shorthand | Origin |
|---|---|---|---|
| N2 | WT | | CGC |
| VL1249 | wwSi15[Pidh-1::idh-1(R156C); unc-119(+) II] | *idh-1neo* | This study |
| EG6699 | *ttTi5605 II; unc-119(ed3) III* | | This study |
| VL1248 | wwSi14[Pidh-1::idh-1(R156H); unc-119(+) II] | | This study |
| VL1250 | wwSi16[Pidh-2::idh-2(R123Q); unc-119(+) II] | | This study |
| VL1243 | wwSi17[Pidh-2::idh-2(R155K); unc-119(+) II] | | This study |
| UP2859 | csls61[RDY-2::GFP, lin-48pro::mRFP] I;jcls1 IV | | Reference 43 |
| VL1409 | wwSi15[Pidh-1::idh-1(R156C); unc-119(+) II]; csls61[RDY-2::GFP,lin-48pro::mRFP] I; jcls1 IV | | This study |

with their promotor regions, were amplified from *C. elegans* genomic DNA using a high-fidelity polymerase. Neomorphic missense mutations were introduced using QuikChange Lightning site-directed mutagenesis kit (Agilent). *C. elegans* strain EG6699 with *mos1* site on chromosome II was used for a direct insertion. 50 animals in the L4/young adult stages were injected with a mix of vectors carrying transgene, Mos1 transposase and selection markers. Injection mix contained 2.5 $\mu$g/ml of pCFJ90 (Pmyo-2::mCherry), 5 $\mu$g/ml pCFJ104 (Pmyo-3::mCherry), 10 $\mu$g/ml pGH8 (Prab-3::mCherry), 50 $\mu$g/ml of pCFJ601 (Peft-3::Mos1 transposase), 10 $\mu$g/ml of pMA122 (Phsp16.41::peel-1), and pCFJ150 with mutated *idh* sequence. Progeny of individual P0 animals were allowed to starve at 25°C and heat shocked at 34°C for 2 h in a water bath. After 4 h of recovery at 20°C WT moving animals without mCherry expression were picked onto individual plates. Resulting lines with full transmittance were verified for transgene integration by PCR.

### *C. elegans* synchronization

Synchronized L1 populations were obtained by treating gravid adult animals with 1% sodium hypochlorite solution buffered with sodium hydroxide. Released embryos were washed with M9 buffer four times and incubated on a rocker for 18–20 h.

### GC-MS metabolomics

Targeted quantification of metabolites by GC-MS was performed as described previously (4). Gravid adult animals were washed three times with filter-sterilized saline (0.9% NaCl). 50 $\mu$l of washed animal pellet were transferred into a FastPrep tube (MP Biomedicals), flash frozen in ethanol/dry ice bath and stored at –80°C. Samples were homogenized in 1 ml of 80% cold methanol with 0.5 ml of acid-washed glass beads (Sigma-Aldrich) using FastPrep24 bead beater (MP-Bio). Supernatant was cleared by centrifugation for 10 min at 10,000*g*. For each sample, 250 $\mu$l of cleared extract were transferred into a glass insert and dried under vacuum. Dry residues were derivatized with 20 $\mu$l of 20 mg/ml methoxyamine hydrochloride (Sigma-Aldrich) in pyridine for 1 h at 37°C. This step was followed by adding 50 $\mu$l of *N*-methyl-*N*-(trimethylsilyl) trifluoroacetamide (Sigma-Aldrich) and a subsequent 3-h incubation at 37°C. After additional 5-h RT incubation, the samples were analyzed on an Agilent single quadrupole mass spectrometer 5977B

coupled with gas chromatograph 7890B. HP-5MS Ultra Inert capillary column (30 m × 0.25 mm × 0.25 $\mu$m) was used with a constant 1 ml/min flow rate of helium gas. Temperature settings were as follows: inlet at 230°C, transfer line at 280°C, MS source at 230°C, and quadrupole at 150°C. A 1 $\mu$l sample was injected in split mode with a 5 ml/min split flow. The initial oven temperature was 80°C, rising to 310°C at a 5°C/min rate. MS parameters included 3 scans/s across a 30–500 m/z range and an electron impact ionization energy of 70 eV. Each metabolite's identification relied on its retention time, a quantifier ion, and two qualifier ions, all manually selected using a reference compound. Peak integration and peak area quantification were executed using Agilent's MassHunter software (v10.1). Blank subtraction and normalization relative to total quantified metabolites were performed using R software.

### Relative quantification of D- and L-2HG

A previously published method (28) was adapted to differentiate the D- and L-enantiomers of 2HG. Initially, 300 $\mu$l of *C. elegans* metabolite extract were dried in glass inserts. 50 $\mu$l of R-(–)-butanol and 5 $\mu$l of 12N hydrochloric acid were then introduced into each insert and heated to 90°C for 3 h. The samples were cooled to RT and extracted with 400 $\mu$l hexane. 250 $\mu$l of the organic phase were dried, the residue was re-suspended in 30 $\mu$l of pyridine and 30 $\mu$l of acetic anhydride and incubated for 1 h at 80°C. The samples were dried once again, resuspended in 60 $\mu$l of hexane, and immediately analyzed by GC-MS. The analytical method settings were identical to the targeted metabolomics method described above, with few modifications. The oven ramp was set from 80°C to 190°C at a rate of 5°C/min and then to 280°C at 15°C/min. D- and L-2HG peaks were quantified using the 173 m/z ion.

### Brood size assay

Animals in the L4 larval stage were singled on 35 mm petri dishes. Every 24 h animals were moved to fresh plates until egg laying ceased. The remaining plates with embryos were incubated at 20°C for 24 h. Subsequently, L1 larvae and unhatched embryos were counted. Brood counts from animals that died or left the plate were excluded. For each biological replicate, data from at least seven animals were collected. The experiment was conducted three times.

### Hatching assay

Approximately 30 synchronized L1 animals were placed on seeded 35 mm NGM agar plates. Animals were incubated at 20°C and allowed to lay eggs. Before eggs start hatching, adults were washed away and ~300 embryos were transferred onto new plates. After 24 h of incubation, hatched larvae and unhatched embryos were counted to determine the rate of embryonic lethality.

### Imaging

Differential interference contrast images were captured with a Zeiss Axioskop fitted with a Leica DFC360 FX camera. Confocal z-stacks were captured with a Leica TCS SP8 confocal microscope. Images were processed using ImageJ.

### RNAi screen

RNAi clones of 2,104 *C. elegans* metabolic genes (30) were cultured in deep 96-well plates in LB (Miller) containing 50 µg/ml ampicillin and grown to stationary phase at 37°C. Cultures were concentrated 20-fold, and 15 µl were plated onto a shallow 96-well plate containing NGM agar supplemented with 64 nM vitamin B12 (adenosylcobalamin), 50 µg/ml ampicillin, and 1 mM IPTG. Plates were dried and stored overnight at 20°C. The next day 15 synchronized L1 animals were added to each well. Plates were screened for strong hatching defects on the 4th and 5th d of incubation at 20°C. The screen was performed three times. All hits were re-tested by performing a hatching assay on 35 mm NGM agar plates.

### Statistical analysis

*P*-values were calculated using unpaired *t* test when comparing two conditions or Tukey's test for multiple pairwise comparisons.

# Supplementary Information

# Acknowledgements

We thank Dr. Ralph DeBerardinis for advice on formate supplementation experiments. This work was funded by grants from the National Institutes of Health DK068429 to AJM Walhout and R35GM136315 to MV Sundaram.

### Author Contributions

O Ponomarova: conceptualization, investigation, methodology, and writing – original draft, review, and editing.
AN Starbard: investigation and methodology.
A Belfi: investigation and methodology.
AV Anderson: investigation and methodology.
MV Sundaram: supervision, funding acquisition, investigation, and writing – original draft.
AJM Walhout: conceptualization, resources, supervision, funding acquisition, project administration, and writing – original draft, review, and editing.

### Conflict of Interest Statement

The authors declare that they have no conflict of interest.

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
