## [Reviewer comments · Life Science Alliance]

Life Science Alliance

idh-1 neomorphic mutation confers sensitivity to vitamin B12 in *Caenorhabditis elegans*

Olga Ponomarova, Alyxandra Starbard, Alexandra Belfi, Amanda Anderson, Meera Sundaram, and Albertha J.M. Walhout
DOI: <https://doi.org/10.26508/lsa.202402924>

Corresponding author(s): Olga Ponomarova, University of New Mexico and Albertha J.M. Walhout, Univ. Massachusetts Medical School

Review Timeline:

Submission Date:	2024-07-02
Editorial Decision:	2024-07-03
Revision Received:	2024-07-04
Accepted:	2024-07-04

Transaction Report:

Please note that the manuscript was reviewed at Review Commons and these reports were taken into account in the decision-making process at Life Science Alliance.

Review
COMMONS

Authors' Response to Reviewers

1. General Statements [optional]

This section is optional. Insert here any general statements you wish to make about the goal of the study or about the reviews.

This section is mandatory. Please insert a point-by-point reply describing the revisions that were already carried out and included in the transferred manuscript.

We thank all three reviewers for their insightful comments. Based on this feedback, we have performed additional experiments, and revised our manuscript. Below, we address each comment and describe the revisions.

Reviewer #1 (Evidence, reproducibility and clarity (Required)):

Summary:

Ponomarova et al. showed that neomorphic *idh-1* mutation results in increased levels of cellular D-2HG. The authors compared the high D-2HG phenotypes by D-2HG dehydrogenase mutant and identified vitamin B12 dependent vulnerability differences. The downregulated gene function of glycine cleavage system involved in one-carbon donor units exacerbates the phenotypes while adding one-carbone donors suppresses the phenotype. They concluded that the *idh-1neo* mutation imposes a dependency on the one-carbon pool. The manuscript is very interesting but I think the manuscript should be modified to be more clear for broad audiences.

Concerns:

The authors mention a number of examples for metabolic changes of D-2HG in the first paragraph of introduction. I think that a metabolic map explaining the changes helps readers to understand the questions proposed by the authors.

Thank you for this suggestion. A figure illustrating the contributing factors in D-2HG metabolism has been added to the manuscript (Figure 1A).

The authors say that D-2HG affects carcinogenesis in many ways, citing previous works. They should say a higher concentration of D-2HG does affect carcinogenesis or not in *dhgd* loss of

function, if they assume the concentration is most important for carcinogenesis.

Thank you for pointing this out. We have added this information in lines 70-72 of the revised manuscript: *“Increased levels of D-2HG caused by the inhibition of D-2-hydroxyglutarate dehydrogenase activity have also been associated with different cancers (PMID: 29339485, PMID: 34296423, PMID: 35007759).”*

Line 110, mode should be read as model, I guess.

Thank you - we have corrected this error.

In Figure 4C, concentrations of formate are shown; 0, 20, 40, 80, 160 mM. Is this correct? the high concentration of substrates changes the osmotic pressure of the medium. Also, high concentration of formic acid is toxic to animals. Considering the concentration of vitamin B12 was 64 nM, I wonder concentration unit of formate is also nM.

We confirm that we supplemented the media with formate in the millimolar range. The highest doses of supplemented formate somewhat slowed the development of P0 animals, but they consistently produced viable progeny. To clarify this we have added the following line to the text on lines 184-187: *“The highest doses of supplemented formate somewhat slowed the development of P0 animals, but restored the survival of *idh-1neo* embryos to wild-type levels on a regular diet of *E. coli* OP50 as well as the diet of RNAi-competent *E. coli* HT115.”*

Additionally, the use of sodium formate ensured that the pH of the media remained unchanged.

I could not understand how embryonic and larval lethality confer the same mechanisms on animal carcinogenesis. Could you explain the logic link between lethal mutation and carcinogenesis. Or do the two phenotypes share only a part of metabolic changes?

Thank you for this suggestion. We have added this in lines 242-246 of the Discussion:

*“While our results have focused on how the neomorphic *idh-1* mutation affects the developing embryo, proliferating cancer cells also have been shown to have increased demand for 1C units, for instance, to synthesize nucleosides (33)(PMID: 24657017). Thus, we can speculate that cancers with mutated *IDH1* may be increasingly sensitive to depletion of the 1C pool, also.”*

Vitamin B12 is an essential substance and deficiency in humans results in severe diseases. Is the lethal phenotype by treatment of *idh-1neo* mutants comparable to humans? Is the concentration of vitamin B12 similar in humans?

The daily dose of human vitamin B12 (cobalamin) in supplements can reach 12.5 µg per kg (PMID: 18606874), while we supplement the media fed to worms with approximately 55 µg

cobalamin per kg (64 nM adenosylcobalamin). No known adverse effects are associated with excessive intake of vitamin B12 by healthy individuals; therefore, no tolerable upper intake level has been set (PMID: 23193625). However, the impact of vitamin B12 on patients with IDH1neo- positive cancers has not been studied.

Reviewer #1 (Significance (Required)):

I think that the manuscript is interesting and may lead an important progress of this field. However, in general, metabolic disorders are difficult to understand for the people outside the speciality. The authors should explain carefully the structure/property, pathways, enzyme functions, and concentration effects of substances of interest.

See above, we hope these edits are sufficient.

Reviewer #2 (Evidence, reproducibility and clarity (Required)):

Increased levels of the metabolite D-2HG (derived from alpha-KG) are associated with multiple disorders. In a previous study, the authors showed that in *C. elegans* dhgd-1 deletion mutants, embryonic lethality resulting from the accumulation of D-2HG in is caused by a lack of ketone bodies. In this study, the authors generated a new model of D-2HG accumulation in *C. elegans*, idh-1neo, in order to further understand how D-2HG exerts its toxic effects in different contexts. This allele mimics mutations found in neomorphic mutations of human IDH1 that lead to abnormal D-2HG production from alpha-KG. Interestingly, the authors find that idh-1neo mutants are distinct from animals lacking the D-2HG dehydrogenase dhgd-1 previously reported. Specifically, while vitamin B12 rescues the embryonic lethality in dhgd-1 deletion animals, it enhances the lethality of idh-1neo animals. Through an elegant genetic screen, and complementation studies with specific metabolites, they provide compelling evidence that this vitamin B12-dependent enhancement is due to depletion of the 1C pool. Specifically, a reverse genetic screen revealed that inactivation of components of the 1 C-producing glycine cleavage system (GCS) results in embryonic lethality in idh-1neo, but not wildtype animals. Complementation studies with specific metabolites show that replenishing C groups is sufficient to reverse embryonic lethality.

This is a very clear, well written paper. Experiments are well controlled and executed, figures are of the highest quality and conclusions are convincing. Prior studies are appropriately referenced.

No additional experiments are required by this reviewer.

Minor points

1) In Figure 2A could authors explain how beta-alanine (increased) is different from alanine (decreased). As a non-specialist this is not clear to me.

Thank you for pointing this out. We added this explanation to the figure legend (lines 521-523).

2) Did the authors test inactivation of the lipoamide dehydrogenase (*dld-1*) has the same effect as the other identified components of the GCS?

The *dld-1* RNAi clone was present in the metabolic library that we screened but was not identified as a "hit." We have added the following in lines 164-168 of the revised manuscript: "*Two other GCS genes, gcs-2 and dld-1 were not identified as 'hits'. gcs-2 is associated with the same reaction as gcs-1, indicating that the latter encodes an active enzyme (30). dld-1 functions in other metabolic processes, particularly in lactate/pyruvate metabolism, and confers embryonic lethality when knocked down in wild type animals (31)*".

Referees cross-commenting

Comments to Reviewer #3:

1/ The authors treat the *idh-1*neo worms with vitamin B12 to reduce 3HP concentrations. The authors should consider conducting experiments to reduce 3HP by other means also. This would help establish a causal relationship between the D-2HG accumulation and observed phenotypes.

The authors show that adding vitamin B12 to the diet of the *idh-1*neo significantly increased their D-2HG levels. Furthermore, *dhgd-1* RNAi drives a further increase in D-2HG in *idh-1*neo animals and led to 100% penetrant embryonic lethality among the F1 generation of *idh-1*neo animals. Together I think this provided strong evidence for a causal relationship between the D-2HG accumulation and observed phenotypes. Further characterizing these phenotypes would be interesting but is beyond the scope of this paper.

4/ The authors should clarify whether it is really vitamin B12 or any other metabolite from the bacteria (like methionine) that is bringing about the phenotypes. Have they tested metabolically inactive bacteria?

the authors show that supplementing B12-treated *idh-1*neo animals with formate (another 1C donor) restored the survival of *idh-1*neo embryos, supporting a role for B12 in depletion of the 1C pool. They also show that suppressing Met/SAM cycle genes in *idh-1*neo prevent 1C depletion and restore availability of 1C units. So the evidence that 1C unit depletion is at the core of the observed phenotypes is pretty convincing

7/ The authors should conduct metabolomic profiling to examine changes in metabolic pathways, including 1C, glycine metabolism, glucose metabolism etc, in *idh-1*neo animals subjected to GCS gene knockdown, and vitamin B12 supplementation.

Not clear how these experiments would add to this story. Open up another line of research

8/ The audience will be limited to the field although the study pertains to an oncometabolite. The study value would have improved if the authors had included cancer cell data. Also, the phenotype studied has not been mechanistically linked to the oncometabolite function, making the study academic in nature.

The interest of this study is that it is being carried out in an organismal context.

Reviewer #2 (Significance (Required)):

As a geneticist with a general interest in metabolomics I find this an elegans study that offers new insight into how IDH-1 and -2 neomorphic mutations affect metabolic rewiring in the context of a whole animal. Although similarities are observed between *idh-1neo* mutants and animals lacking the D-2HG dehydrogenase *dhgd-1*, both of which have increased levels of the metabolite D-2HG, specific metabolic differences are observed. The identification of 1C unit deficiency as a driver of lethality in *idh-1neo* mutants is highly significant given the central importance of 1C metabolism. This study should therefore be of interest to a wide audience.

Reviewer #3 (Evidence, reproducibility and clarity (Required)):

Ponomarova et al presents a short follow up of their previous study to elucidate the role of a oncogenic variant of *idh-1* that increases the 3HP levels, similar to the δ *dhgd-1* mutant. Using a combination of metabolomics and genetics, they show that the defect in *idh-1neo* worms on high vitamin B12 diet is the draining of the 1C pool, distinct from the mechanisms of lethality observed in the δ *dhgd-1* mutant. While the findings are interesting, there is a lack of mechanistic understanding of the basis of the phenotype observed. Moreover, the authors do not establish the link between the oncometabolite, that should support uncontrolled cell division, with the observed phenotype. Some control experiments are missing and should be included in the revised manuscript. there could be many other The comments on the manuscript are as follows, in no particular order:

1. The authors treat the *idh-1neo* worms with vitamin B12 to reduce 3HP concentrations. The authors should consider conducting experiments to reduce 3HP by other means also. This would help establish a causal relationship between the D-2HG accumulation and observed phenotypes.

To further examine the link between 3HP and *idh-1neo* embryonic lethality, we targeted *hphd-1* by RNAi, which increases 3HP levels (Ponomarova et al., 2023). *Hphd-1* knockdown did not induce lethality in the wild-type or exacerbate lethality in *idh-1neo* animals (Figure S3), further demonstrating that lack of 3HP degradation is not linked to this phenotype (lines 143-145).

Also, see cross-comments from Reviewer #2 above.

2. The authors should investigate the functional impact of HPHD-1 inhibition on 3-hydroxypropionate levels and D-2HG accumulation by RNAi knockdown of HPHD-1 in *idh-1neo* animals.

We have now performed the suggested experiment please see response to comment 1 above.

3. The authors do not clearly mention clearly which diet in some of their experiments. This is important since the two diets used (OP50 and HT115) differ in their vitamin B12 content, and thus could have different consequences.

We added this information in figures, figure legends, and lines 259-260 of the revised manuscript.

4. The authors should clarify whether it is really vitamin B12 or any other metabolite from the bacteria (like methionine) that is bringing about the phenotypes. Have they tested metabolically inactive bacteria?

The reviewer correctly points out that bacterial metabolism may play a role in the effects exerted by vitamin B12. We have not tested metabolically inactivated bacteria, however, our RNAi experiments (Figure 4E) demonstrate that supplemented vitamin B12 acts through the Met/SAM cycle in *idh-1neo* animals. Please also see cross-comments from Reviewer #2.

5. The authors consistently use 64 nM of Vitamin B12. Will the *hphd-1* mutant and the *idh-1neo* mutant have different vitamin B12 thresholds for the observed phenotypes?

Thank you for raising this interesting point. While 64 nM vitamin B12 virtually eliminates 3HP accumulation in *idh-1* animals (Figure 2D), we have not tested if this dose is sufficient to eliminate 3HP accumulation in *hphd-1* mutant. However, potential differences in 3HP levels in *idh-1neo* and *hphd-1* animals treated with vitamin B12 would not contradict our conclusion that 3HP is not the cause of embryonic lethality in *idh-1neo* mutant animals.

6. Figure 3b: HT115 has inherently high levels of vitamin B12 so the RNAi effect of genes should be seen on the OP50 diet supplemented with B12.

Despite reports of elevated B12 levels in *E. coli* HT115, vitamin B12-induced embryonic lethality of *idh-1neo* on a diet of OP50 is more severe than on a diet of HT115 bacteria (Figure 4C).

Therefore, it may be harder to quantify synthetic lethal interaction of *idh1-neo* with GCS RNAi knockdown using OP50 strains (which would need to be created).

7. The authors should conduct metabolomic profiling to examine changes in metabolic pathways, including 1C, glycine metabolism, glucose metabolism etc, in *idh-1neo* animals subjected to GCS gene knockdown, and vitamin B12 supplementation.

While these results would be interesting and further our understanding of metabolic changes that occur in *idh-1neo* mutant animals we think they are beyond the scope of the manuscript. Also, please see cross-comments from Reviewer #2.

8. Perform rescue experiments using different one-carbon donors (e.g., formate, serine) to restore embryonic viability in *idh-1neo* mutants under conditions of vitamin B12-induced stress. Quantify the efficacy of these interventions using developmental assays.

In addition to formate rescue experiments (Figure 4C), we supplemented *idh-1neo* animals with serine (Figure 4D and S7). Similar to formate, serine supplementation resulted in the rescue of *idh-1neo* embryonic lethality on an *E. coli* OP50 diet (lines 187-189). The lack of rescue on an HT115 diet could be due to HT115 bacteria containing more glycine (Gao et al., 2017), which might limit the efficiency of serine conversion to glycine needed for 1C unit production.

9. Provide experimental evidence to show that *idh-1neo* animals possess an alternative source of energy.

We have previously found that diminished production of ketone bodies in $\Delta dhgd-1$ mutants causes embryonic lethality that can be rescued by exogenous supplementation of ketone body 3-hydroxybutyrate (Ponomarova et al., 2023). In contrast to *dhgd-1* mutants, *idh-1neo* embryonic lethality fails to respond to supplemented 3-hydroxybutyrate (Figure S4), indicating the lethality associated with the *idh-1neo* mutation is caused by a different mechanism, i.e., a depletion in 1C-units.

10. The authors use vitamin B12 to inhibit the shunt pathway (line 127). They should explore alternate strategies to do the same, like gene knockdown.

Please see our response to comment 1 above where we discuss RNAi knock-down of the shunt pathway gene, *hphd-1*.

11. It is not clear why the authors did not follow up with the other phenotypes of the *idh-1neo* that were visible without the Vitamin B12 supplementation. They should follow up with this and also other phenotypes to explore the broader physiological consequences of D-2HG accumulation.

We agree that the other physiological consequences of D-2HG accumulation are interesting, and we plan to investigate them in our future studies.

12. The authors should include control experiments without supplementation of vitamin B12, ketone bodies etc. in each of their figures.

We thank the reviewer for this suggestion. We have added these data (Figures S5, 6, 7, and 8).

13. The authors posit that the *idh-1neo* depletes the 1C pool leading to the observed lethality. So, when they supply formate to replenish it, they rescue the lethality of the B12-treated worms. Similar results are obtained by knocking down the enzymes. So where are the 1C units going? Understanding this will provide the much-needed mechanistic understanding to this study.

We appreciate this insightful comment and expand our discussion to elaborate on this issue (lines 224-227). “We propose that a lack of 1C units in *idh-1neo* can impede pyrimidine biosynthesis via thymidylate synthase *tyms-1*, which uses 1C units to generate dTMP. Supporting this hypothesis, RNAi of *tyms-1* causes embryonic lethality (36-38).”

14. It may be important to measure the D-2HG levels in the mitochondria vs the cytosol.

While this is an interesting point, we think that this line of inquiry is beyond the scope of this work (and is technically challenging).

15. The *idh-1neo* is an oncometabolite. The authors do not show any data to indicate whether this mutant has any defect in cell division/cell cycle in the somatic tissue or germline.

In this study we primarily focused on the molecular changes in the metabolic network that occur in *idh-1neo* mutant animals, which we think is an important advance in understanding the basis for how this mutation affects IDH function. Additional phenotypic outcomes of these perturbed metabolic processes will be the basis of future studies.

Reviewer #3 (Significance (Required)):

The audience will be limited to the field although the study pertains to an oncometabolite. The study value would have improved if the authors had included cancer cell data. Also, the phenotype studied has not been mechanistically linked to the oncometabolite function, making the study academic in nature.

While we agree that the link between *idh-1neo*, 2HG production and oncometabolite function has not been directly shown we think that our study adds important molecular understanding of metabolic changes that occur in relation to *idh-1neo* function which are important for future studies of how this mutation affects carcinogenesis. Also, please see cross-comments from Reviewer #2.

In addition, we specified statistical significance in Figure 2, described statistical tests used (lines 361-363) and corrected a few grammatical errors throughout the text.

July 3, 2024

RE: Life Science Alliance Manuscript #LSA-2024-02924-T

Olga Ponomarova
University of New Mexico School of Medicine
Albuquerque 87131

Dear Dr. Ponomarova,

Thank you for submitting your revised manuscript entitled "idh-1 neomorphic mutation confers sensitivity to vitamin B12 in *Caenorhabditis elegans*". We would be happy to publish your paper in Life Science Alliance pending final revisions necessary to meet our formatting guidelines.

- please be sure that the authorship listing and order is correct
- please upload your main manuscript text as an editable doc file
- please upload all figure files as individual ones, including the supplementary figure files
- please add a Running Title and a Summary Blurb/Alternate Abstract to our system
- please add ORCID ID for the corresponding author -- you should have received instructions on how to do so
- please add a Category for your manuscript in our system
- please add the Twitter handle of your host institute/organization as well as your own or/and one of the authors in our system
- titles in the system and manuscript file must match
- please label the table on pg. 9 as Table 1
- please add an Author Contributions section to your main manuscript text, and our system
- please add a Conflict of Interest statement to your main manuscript text
- please upload your Tables in editable .doc or Excel format
- please add callouts for Figures 1; S1A-B; S6A-B and S7A-B to your main manuscript text

LSA now encourages authors to provide a 30-60 second video where the study is briefly explained. We will use these videos on social media to promote the published paper and the presenting author (for examples, see <https://docs.google.com/document/d/1-UWCfbE4pGcDdcgzcmiuJl2XMBJnxKYeqRvLLrLS08s/edit?usp=sharing>). Corresponding or first-authors are welcome to submit the video. Please submit only one video per manuscript. The video can be emailed to contact@life-science-alliance.org

A. FINAL FILES:

B. MANUSCRIPT ORGANIZATION AND FORMATTING:

Sincerely,

July 4, 2024

RE: Life Science Alliance Manuscript #LSA-2024-02924-TR

Olga Ponomarova
University of New Mexico
Albuquerque 87131

Dear Dr. Ponomarova,

Thank you for submitting your Research Article entitled "idh-1 neomorphic mutation confers sensitivity to vitamin B12 in *Caenorhabditis elegans*". It is a pleasure to let you know that your manuscript is now accepted for publication in Life Science Alliance. Congratulations on this interesting work.

DISTRIBUTION OF MATERIALS:

Again, congratulations on a very nice paper. I hope you found the review process to be constructive and are pleased with how the manuscript was handled editorially. We look forward to future exciting submissions from your lab.

Sincerely,
